# Spatial profiling of non-small cell lung cancer provides insights into tumorigenesis and immunotherapy response

Joon Kim [1,2,5], Seung Hyun Yong[2,5], Gyuho Jang [1], Yumin Kim[1], Raekil Park [1], Hyun-Hee Koh [3], Sehui Kim[3,4] ✉, Chang-Myung Oh [1] ✉ & Sang Hoon Lee[2] ✉

Lung cancer is the second most common cancer worldwide and a leading cause of cancer-related deaths. Despite advances in targeted therapy and immunotherapy, the prognosis remains unfavorable, especially in metastatic cases. This study aims to identify molecular changes in non-small cell lung cancer (NSCLC) patients based on their response to treatment. Using tumor and matched immune cell rich peritumoral tissues, we perform a retrospective, comprehensive spatial transcriptomic analysis of a proven malignant NSCLC sample treated with immune checkpoint inhibitor (ICI). In addition to T cells, other immune cell types, such as B cells and macrophages, were also activated in responders to ICI treatment. In particular, B cells and B cell-mediated immunity pathways are consistently found to be activated. Analysis of the histologic subgroup (lung squamous cell carcinoma, LUSC; lung adenocarcinoma, LUAD) of NSCLC also confirms activation of B cell mediated immunity. Analysis of B cell subtypes shows that B cell subtypes were more activated in immune cell-rich tissues near tumor tissue. Furthermore, increased expression of B cell immunity-related genes is associated with better prognosis. These findings provide insight into predicting ICI treatment responses and identifying appropriate candidates for immunotherapy in NSCLC patients.

Lung cancer is the leading cause of cancer incidence and mortality worldwide, accounting for an estimated 2 million diagnoses and 1.8 million deaths[1]. NSCLC is the predominant subtype of lung cancer, affecting approximately 85% of patients[1,2]. Although many therapeutic strategies such as surgery, radiation, chemotherapy, and molecular targeted therapy are available for the treatment of NSCLC, the prognosis of lung cancer remains poor, with a 5-year survival rate of only 23%[3]. In addition, many patients who were initially considered to have early-stage disease and who underwent surgery are susceptible to distant metastases or local recurrence[2]. Considering the nature of NSCLC, patients may require systemic treatment during the treatment process. Recent advances in cancer treatment have focused on developing drugs that target the interaction of the immune system with tumors and have robust systemic anticarcinogenic effects.

Immunotherapy, or biological treatment, boosts the innate human immune system's anti-cancer defenses[2].

Over the last 10 years, ICI have shown significant survival benefits in patients with advanced NSCLC[4,5]. According to the American Lung Association (State of Lung Cancer, 2021), 5-year survival rates for NSCLC increased from 14% to 23.7%[5]. However, despite advances in overall survival, ICI-treated patients with NSCLC still showed an overall poor outcome, and newly adapted treatments also brought about unique toxicities or related problems. Some patients suffered from immune-related toxicities, such as enterocolitis, endocrinopathies, hepatitis, and myocarditis[6]. Furthermore, some patients failed to respond to ICI therapy, which could not be confirmed. This failure may be associated with the development of primary or acquired resistance[7]. However, there remains a lack of knowledge regarding patients' responses to immunotherapy[7].

[1]Department of Biomedical Science and Engineering, Gwangju Institute of Science and Technology, Gwangju, Korea. [2]Division of Pulmonary and Critical Care Medicine, Department of Internal Medicine, Severance Hospital, Yonsei University College of Medicine, Seoul, Republic of Korea. [3]Department of Pathology, Severance Hospital, Yonsei University College of Medicine, Seoul, Republic of Korea. [4]Department of Pathology, Korea University Guro Hospital, Seoul, Republic of Korea. [5]These authors contributed equally: Joon Kim, Seung Hyun Yong. ✉e-mail: sulparangi@naver.com; cmoh@gist.ac.kr; cloud9@yuhs.ac

Recently, single-cell sequencing technologies have revealed previously unknown heterogeneity in cancer cells and the surrounding cellular environment, including cells responsible for immune reactions. However, single-cell sequencing techniques require tissue dissociation, which results in a loss of spatial context. Spatial transcriptomics has rapidly emerged as a solution, allowing the construction of tissue atlases and the characterization of spatiotemporal heterogeneity in cancer[8,9].

In this study, we sought to identify differences in transcriptome expression profiling in response to treatment in patients receiving ICI treatment for metastatic NSCLC. We performed spatial transcriptome analysis of lung tissues from NSCLC patients who are candidates for immune checkpoint inhibitors and classified the lung tissues into tumor tissues and immune cell-rich tissues near the tumor. We aimed to identify molecular and functional changes in both tumor and peritumoral immune cell-rich tissues in response to ICI treatment in NSCLC patients. These comprehensive analyses provide valuable insights into the molecular landscape of NSCLC and contribute to our understanding of immunotherapy response mechanisms.

## Results
### Patient characteristics
Spatial transcriptome analysis was conducted on the lung tissues obtained from 18 patients with stage IV NSCLC who were candidates for immune checkpoint blockade. The clinical characteristics of the patients are summarized in Table 1.

The response to immunotherapy was evaluated following RECIST 1.1 criteria. This cohort comprised 7 responders and 11 non-responders. Patients were defined as responders if the duration of treatment response to ICI was greater than or equal to 180 days and non-responders if the duration of treatment response was less than 180 days. There were no significant differences in age, BMI, smoking history, or PD-L1 expression between the responder and non-responder groups. The majority of the NSCLC subtypes in both groups were squamous cells, followed by adenocarcinomas. Hypertension was the most common comorbidity in both groups. In the response group, three patients were treated with pembrolizumab, and four patients were treated with nivolumab. In the non-responder group, six patients were treated with pembrolizumab, while five patients were treated with nivolumab. In the responder group, the most common site of metastasis was the contralateral lung, followed by the pleura, while in the non-responders, the most common site of metastasis was the pleura.

### Identification of NSCLC tumor and immune samples
In collaboration with a pathologist, we meticulously identified tissue regions corresponding to tumor and immune cells, allowing for a focused analysis of spatial profiling within these regions (Fig. 1a, Supplement Fig. 1a).

ESTIMATE analysis was performed on the entire NSCLC samples and histologic subtypes of NSCLC (LUSC, LUAD) to determine whether the tissue regions were appropriately divided into tumor and immune samples. The stromal score, immune score, and ESTIMATE score were all elevated in immune samples of both entire NSCLC samples and each histological subtype compared to tumor samples, confirming the distinct classification of tumor and immune tissues (Supplementary Fig. 1b–d, Supplementary Data 1).

### Transcriptome expression profiling between NSCLC tumor and immune samples
Prior to the analysis of transcriptome expression profiling for NSCLC immune checkpoint blockade responsiveness, we performed an analysis to identify the difference in transcriptome expression profiling between tumor and immune tissues (Supplementary Fig. 2).

We identified DEGs with a $P_{adj} \leq 0.05$ between tumor and immune tissues in entire NSCLC samples and identified 2895 upregulated and 3909 downregulated DEGs. DEGs were also identified between tumor and immune samples of NSCLC histologic subtypes, with 2262 upregulated and

2998 downregulated DEGs in LUSC and 435 upregulated and 379 downregulated DEGs in LUAD (Supplementary Fig. 2a–c, Supplementary Data 2).

Next, GSEA was performed using MSigDB Hallmark, GO BP, KEGG, and cell type signature gene sets to identify functional changes between tumor and immune samples in NSCLC samples (Supplementary Fig. 2d–g, Supplementary Data 2). GSEA using Hallmark terms confirmed that metabolism, proliferation pathways, and most signaling pathways were significantly positively enriched in tumor samples, whereas the immune pathway was negatively enriched (Supplementary Fig. 2d). Similar results were found in GSEA using GO BP and KEGG terms, and in particular, GSEA using cell type signature gene sets confirmed that T cells, B cells, and immune cells such as neutrophils, macrophages, and NK cells were negatively enriched in tumor samples (Supplementary Fig. 2g).

### Transcriptome expression profiling of NSCLC immune checkpoint blockade responsiveness
In collaboration with a pathologist, we meticulously identified tissue regions corresponding to tumor and immune cells, allowing for a focused analysis of spatial profiling within these regions (Fig. 1a).

To investigate the differences between responders and non-responders to ICI treatment, we identified differentially expressed genes (DEGs) between responders and non-responders in the entire NSCLC samples and in tumor and immune samples. In entire NSCLC samples, 302 upregulated and 362 downregulated DEGs between responders and non-responders were identified with $P_{nom} \leq 0.05$. Similarly, DEGs between responders and non-responders were identified in NSCLC tumor and immune samples, with 391 upregulated and 486 downregulated DEGs identified in tumor samples and 130 upregulated and 262 downregulated DEGs identified in immune samples (Fig. 1b–d, Supplementary Data 3).

Next, GSEA was performed using MSigDB Hallmark, GO BP, KEGG terms, and cell type signatures to identify functional differences between responders and non-responders to ICI treatment (Fig. 1e–h, Supplementary Data 3). GSEA using Hallmark and GO BP terms showed that pathways related to glucose metabolism and proliferation and ribosomal pathways involved in protein synthesis were positively enriched in the ICI responders compared to the non-responders, and in terms of immune-related pathways, the antigen processing and presentation pathways were positively enriched in the responders (Fig. 1e, f). In GSEA using KEGG terms, similar to the analysis using Hallmark and GO BP terms, we found that translation, ribosome biogenesis, and peptide biosynthesis pathways related to anabolism and glucose metabolism-related pathways were positively enriched in the immune checkpoint inhibitor responders than in the non-responders, and in addition, activation of immune response and B cell-mediated immunity-related pathways were positively enriched in the responders, specifically in NSCLC immune samples (Fig. 1g). GSEA performed using a cell type signature gene set to identify immune cell functions showed positive enrichment of both CD4 and CD8 memory effector T cells, as well as B cells, in the NSCLC immune checkpoint inhibitor responders, and innate cell immunity-related immune cells such as monocytes and macrophages were positively enriched in the responders, specifically in NSCLC immune samples (Fig. 1h).

### Transcriptome expression profiling of LUSC immune checkpoint blockade responsiveness
Subtype analysis was performed on LUSC and LUAD samples to identify differences between responders and non-responders to immune checkpoint inhibitor treatment in each histologic subtype of NSCLC. In the LUSC samples, we identified DEGs with a nominal $P$ value $\leq 0.05$ between immune checkpoint inhibitor responders and non-responders, with 232 upregulated and 275 downregulated genes in the entire LUSC sample. Similarly, we identified 330 upregulated and 470 downregulated genes in LUSC tumor samples and 102 upregulated and 181 downregulated genes in LUSC immune samples (Fig. 2a–c, Supplementary Data 4).

## Table 1 | Patient demographics and clinical characteristics

| Patients characteristics, total n | Responder (7) | Non-responder (11) |
|---|---|---|
| Age, y, median ± SD | 61.0 ± 10.6 | 66.0 ± 9.3 |
| *Sex, n (%)* | | |
| Male | 7 (100.0) | 8 (72.7) |
| Female | 0 (0.0) | 3 (27.3) |
| BMI, mean ± SD | 25.6 ± 4.9 | 21.3 ± 3.9 |
| Smoking history, PYR, mean ± SD | 34.6 ± 16.6 | 22.0 ± 17.6 |
| Total treatment duration, median ± SD | 719 ± 105.9 | 91 ± 43.6 |
| *Patients, total n (%)* | | |
| With adenocarcinoma | 2(28.6) | 4(36.4) |
| With squamous cell carcinoma | 4(57.2) | 7(53.6) |
| With another type of non-small cell lung cancer | 1(14.2) | 0(0.0) |
| *Patients' comorbidities, n* | | |
| Hypertension | 4 | 5 |
| Diabetes mellitus | 2 | 3 |
| Chronic kidney disease | 1 | 1 |
| Coronary artery disease | 2 | 2 |
| Cerebrovascular disease | 1 | 2 |
| Chronic lung infection (e.g., Fungal, NTM) | 1 | 0 |
| History of other malignancy | 2 | 3 |
| *Characteristics by type of regimen, total n (%)* | | |
| Pembrolizumab | 3(42.9) | 6(54.5) |
| Nivolumab | 4(47.1) | 5(45.5) |
| *Stage on diagnosis* | | |
| Stage I | 0 | 0 |
| Stage II | 2 | 1 |
| Stage III | 4 | 3 |
| Stage IV | 2 | 7 |
| *PD-L1 expression by percentage, mean ± SD* | | |
| SP263 | 39.3 ± 27.5 | 31.25 ± 29.5 |
| IHC 22C3 | 39.2 ± 36.4 | 39.1 ± 30.1 |
| *Metastatic lesion, n* | | |
| Contralateral lung | 4 | 6 |
| Brain | 1 | 5 |
| Bone | 1 | 5 |
| Pleura | 3 | 8 |
| Other | 1 | 7 |

Abbreviation: *BMI* body mass index, *NTM* non-tuberculous mycobacterium, *PYR* pack-years, *SD* standard deviation.

Next, GSEA was performed to identify functional differences between immune checkpoint inhibitor responders and non-responders in LUSC (Fig. 2d–g, Supplementary Data 4). GSEA using MSigDB Hallmark and GO BP terms revealed positive enrichment of pathways related to glucose metabolism and proliferation, ribosome, antigen processing, and presentation in the responders as well as in the whole NSCLC samples (Fig. 2d, e). GSEA using KEGG terms also confirmed the positive enrichment of anabolism-related pathways such as translation, ribosome biogenesis, peptide, amide biosynthesis, and other immune-related pathways, and the positive enrichment of antigen processing and presentation-related pathways (Fig. 2f). GSEA using the cell type signature gene set identified positive enrichment of CD4 memory effector T cells, B cells, and macrophages specific to immune samples (Fig. 2g).

### Transcriptome expression profiling of LUAD immune checkpoint blockade responsiveness

Next, we identified DEGs with a $P_{nom} \leq 0.05$ between immune checkpoint inhibitor responders and non-responders in the LUAD samples. In the entire LUAD samples, we identified 178 upregulated and 179 down-regulated DEGs, and 293 upregulated and 256 downregulated DEGs in LUAD tumors. And, in the LUAD immune sample, we identified 71 upregulated and 195 downregulated DEGs (Fig. 3a–c, Supplementary Data 5).

In LUAD, as in the entire NSCLC and LUSC samples, we performed functional analysis to identify differences between ICI responders and non-responders (Fig. 3d–g, Supplementary Data 5). GSEA using MSigDB Hallmark terms confirmed that LUAD ICI responders were positively enriched for interferon-gamma response compared to non-responders. In the metabolism-related pathways, only fatty acid metabolism was positively enriched, while proliferation pathways were mostly negatively enriched. GSEA using GO BP terms revealed positive enrichment of the translation-related ribosome pathway, similar to the whole NSCLC and LUSC samples. The glycolysis/gluconeogenesis pathway was found to be positively enriched only in responders of LUAD tumor samples. Analysis using KEGG terms showed positive enrichment of the translation and amide biosynthesis pathways in entire LUAD samples and LUAD immune samples, as well as the B cell-mediated immunity-related pathway and phagocytosis-related pathway in entire NSCLC samples and LUSC samples. GSEA, using the cell type signature gene set, confirmed the positive enrichment of macrophages.

### Identifying modules associated with ICI treatment response using NMF

To identify gene modules associated with ICI treatment response, we performed NMF using responder-specific DEGs and tumor-specific DEGs (Fig. 4a–l, Supplementary Data 6). We named the responder's DEGs among NSCLC tumor samples as a tumor response signature and used it to identify whether NSCLC tumor samples were divided into clusters (Fig. 4a). Cluster 2 was identified as a non-responder-specific cluster, and we found that the enrichment score of cluster 2 genes was higher in non-responders than responders in the entire NSCLC tumor samples, but did not reach significance ($P = 0.074$) (Fig. 4b, c). We performed ORA for functional analysis of cluster 2 genes and found that differentiating basal cells were significantly enriched (Fig. 4d).

We also named the DEGs of responders among NSCLC immune samples as immune response signatures and used them to check whether NSCLC immune samples were divided into clusters (Fig. 4e). We identified cluster 1 as a responder-specific cluster and found that cluster 1 genes in the entire NSCLC immune samples had significantly higher enrichment scores in responders than non-responders ($P = 0.035$) (Fig. 4f, g). We performed ORA on cluster 2 genes and found enrichment of CD4 T cells, B cells, as well as dendritic cells and monocytes (Fig. 4h).

Lastly, we named the DEGs of NSCLC tumor and immune samples as tumorigenesis signatures and used them to check whether NSCLC tumor samples were divided into clusters (Fig. 4i). We found that cluster 1 is a cluster that is relatively specific to non-responders, and we found that cluster 1 genes had significantly higher enrichment scores in non-responders than in responders across all NSCLC tumor samples ($P = 0.035$) (Fig. 4j, k). ORA on cluster 1 genes revealed enrichment of hypoxia, p53 pathway, and others, confirming that these pathways are associated with the non-responder subtype (Fig. 4l).

### Identifying modules associated with ICI treatment response using WGCNA

WGCNA was performed as an additional method in addition to NMF to identify gene modules associated with immune checkpoint inhibitor treatment responses. In the entire NSCLC sample, 13 modules were identified, with a median of 594 genes per module. In NSCLC tumor samples, 129 modules were identified, with a median of 59 genes per module. In NSCLC

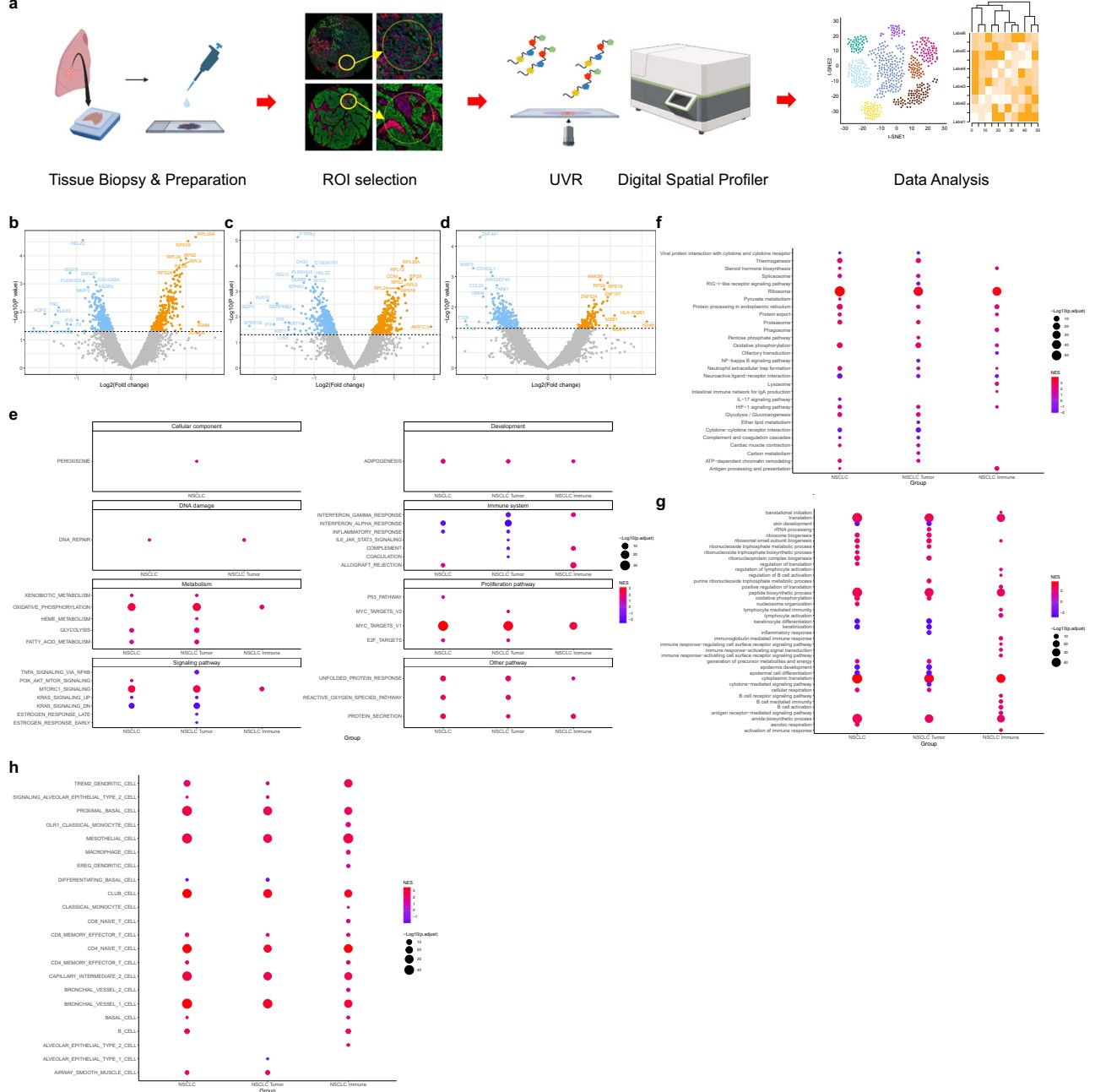

**Fig. 1 | Expression profiling and molecular characteristics of NSCLC transcriptome by response. a** Study design and analysis flow diagram. **b–d** Volcano plot of DEGs between responder and non-responder of **b** all NSCLC (*n* = 18), **c** tumor, and **d** immune samples. Only genes with a nominal *p*-value ≤ 0.05 were considered DEGs. **e** Dotplot showing GSEA results using MSigDB hallmark gene sets between NSCLC responder and non-responder samples. **f** Dotplot showing top 20 GSEA results using MSigDB KEGG gene sets between NSCLC responder and non-responder samples. **g** Dotplot showing top 20 GSEA results using MSigDB GO BP gene sets between NSCLC responder and non-responder samples. **h** Dotplot showing GSEA results using MSigDB cell type signature gene sets between NSCLC responder and non-responder samples. DEGs differentially expressed genes, GSEA gene set enrichment analysis, MSigDB molecular signature database, GO BP gene ontology biologic process, NES normalized enrichment score.

immune samples, 175 modules were identified, with a median of 72 genes per module (Supplementary Fig. 3a–c).

In the entire NSCLC samples, none of the modules associated with immune checkpoint inhibitor treatment responses were significant with a *P* value ≤ 0.05, whereas eight significant modules were identified in the NSCLC tumor samples and two in the immune samples. To determine whether these modules were enriched differently between responders and non-responders, we performed ssGSEA and found that among the modules identified in NSCLC tumor samples, salmon1 (*P* = 0.021), dark turquoise (*P* = 0.014), and dark slate blue module (*P* = 0.011) were found to have

significantly different enrichment scores in responders and non-responders, with the salmon1 module having a higher enrichment score in non-responders and the dark turquoise and dark slate blue modules having a higher enrichment score in responders (Fig. 5a, b, Supplementary Data 7). To determine the function of each of these three modules, we performed ORA on each of the modular genes and found that neutrophils, monocytes, macrophages, and CD8 T cells in the dark turquoise module, O-glycan and several amino acid metabolism-related pathways in the salmon1 module, and IL6/JAK/STAT3 pathway in the dark slate blue module were enriched (Fig. 5c–e, Supplementary Data 7).

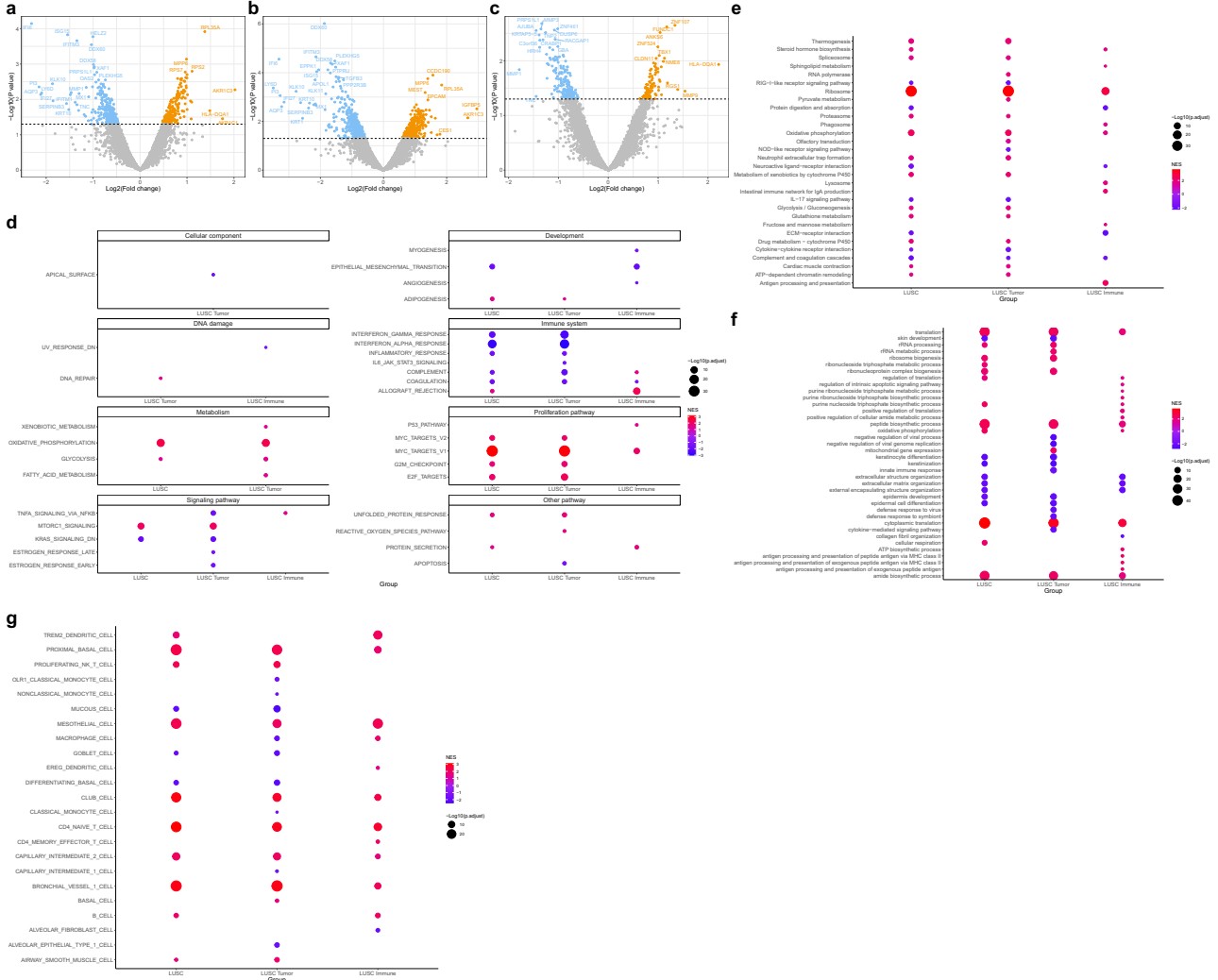

**Fig. 2 | Expression profiling and molecular characteristics of LUSC transcriptome by response. a–c** Volcano plot of DEGs between responder and non-responder of **a** all LUSC (*n* = 11), **b** tumor, and **c** immune samples. Only genes with a nominal *P*-value ≦ 0.05 were considered DEGs. **d** Dotplot showing GSEA results using MSigDB hallmark gene sets between LUSC responder and non-responder samples. **e** Dotplot showing top 20 GSEA results using MSigDB KEGG gene sets between LUSC responder and non-responder samples. **f** Dotplot showing top 20 GSEA results using MSigDB GO BP gene sets between LUSC responder and non-responder samples. **g** Dotplot showing GSEA results using MSigDB cell type signature gene sets between LUSC responder and non-responder samples. DEGs differentially expressed genes, GSEA gene set enrichment analysis, MSigDB molecular signature database, GO BP gene ontology biologic process, NES normalized enrichment score.

## Identifying associations between immune cell subtypes and ICI treatment response

The above analyses have repeatedly shown that B cells and macrophages were associated with ICI treatment responses, and we conducted a literature review to identify markers associated with B cell and macrophage subtypes for further analysis (Supplementary Data 8).

We then identified enrichment scores for each immune cell subtype by GSEA in the entire NSCLC, NSCLC tumor, and immune samples to determine whether the function of each immune cell subtype was upregulated or downregulated in ICI responders and non-responders. In the ICI responders, we found positive enrichment scores for all B cell populations and the M1 macrophage subtype and negative enrichment scores for the M2 macrophage subtype. These changes were more significant in immune samples than in NSCLC tumor samples (Fig. 6a, Supplementary Data 8).

To confirm whether B cell and M1 macrophage subtypes were enriched in other data, we performed ssGSEA on the validation dataset (GSE126044[10], GSE135222[11]) to check the enrichment scores of B cell and macrophage subtypes. In the validation dataset, we found that the enrichment score of all B cell subtypes in ICI responders was higher than in non-responders, with activated B cells and switched memory B cells having significantly higher enrichment scores in responders than non-responders (Fig. 6b, Supplementary Data 8).

In the above analysis, we found that B cell and M1 macrophage function varied in response to immune checkpoint inhibitor treatment, and to determine if this actually affected patient survival, we performed Kaplan–Meier survival analysis and found that median survival of the high B cell immunity group (749 days) was higher than that of the intermediate or low B cell immunity group (171 and 239 days) in the Severance cohort, although it did not reach statistical significance (*P* = 0.057), and in the validation dataset, median survival was significantly increased in the high B cell immunity group (189 days) than the low B cell immunity group (89 days).

## Discussion

In this study, we performed spatial transcriptome analysis on lung samples obtained from patients with metastatic NSCLC to investigate the relationship between gene expression characteristics and ICI treatment response. Our analysis focused on comparing the gene expression patterns, tumor

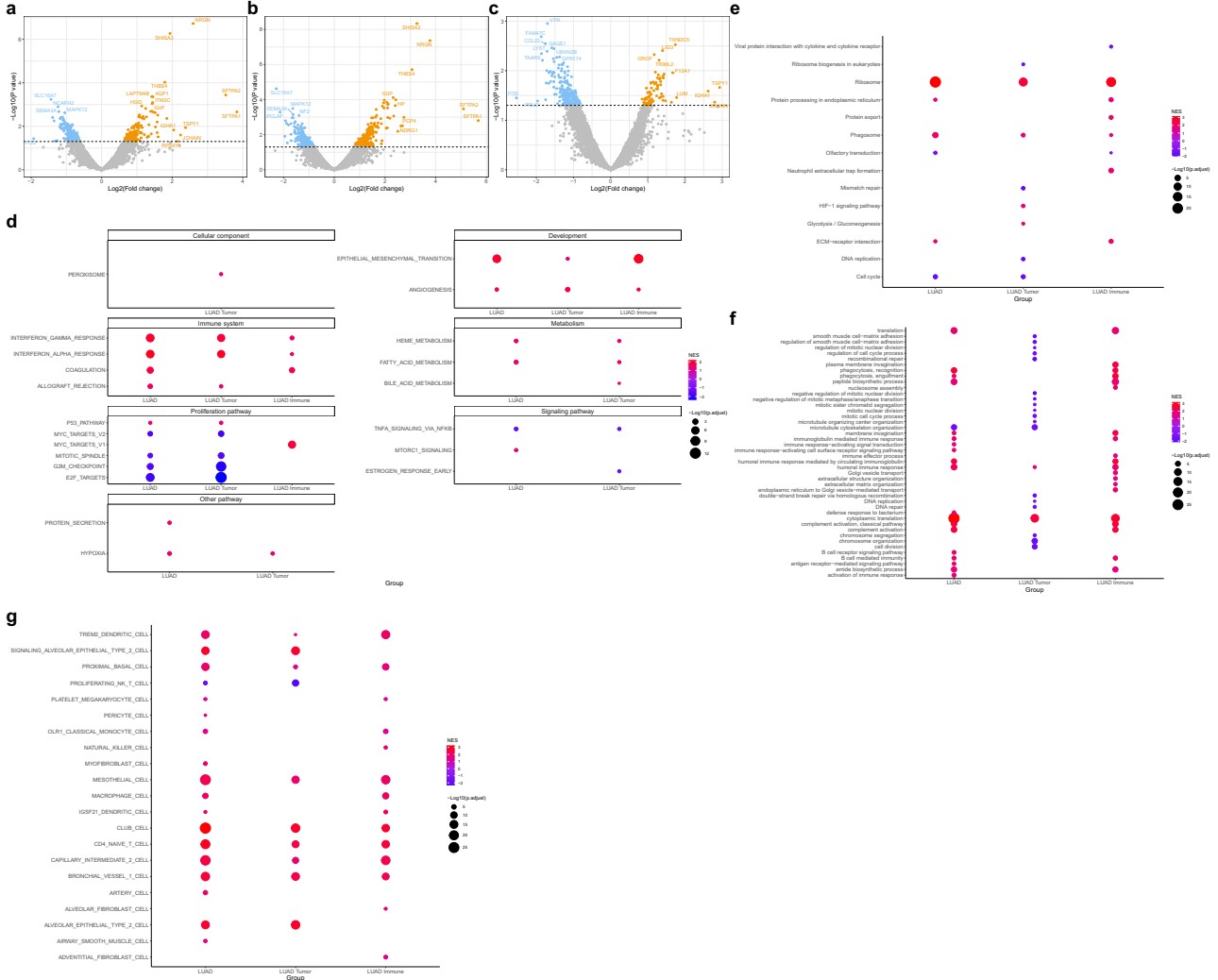

**Fig. 3 | Expression profiling and molecular characteristics of LUAD transcriptome by response. a–c** Volcano plot of DEGs between responder and non-responder of **a** all LUAD (n = 6), **b** tumor, and **c** immune samples. Only genes with a nominal P-value ≦ 0.05 were considered DEGs. **d** Dotplot showing GSEA results using MSigDB hallmark gene sets between LUAD responder and non-responder samples. **e** Dotplot showing top 20 GSEA results using MSigDB KEGG gene sets between LUAD responder and non-responder samples. **f** Dotplot showing top 20 GSEA results using MSigDB GO BP gene sets between LUAD responder and non-responder samples. **g** Dotplot showing GSEA results using MSigDB cell type signature gene sets between LUAD responder and non-responder samples. DEGs differentially expressed genes, GSEA gene set enrichment analysis, MSigDB molecular signature database, GO BP gene ontology biologic process, NES normalized enrichment score.

microenvironment, and pathway enrichment between responders and non-responders, as well as investigating the specific features of immunotherapy responsiveness in the histological subtypes of NSCLC, namely LUSC and LUAD.

We identified gene expression patterns in response to treatment in NSCLC samples and found that translation-related genes and some immune-related genes were upregulated, while some other immune-related genes were downregulated in ICI responders. GSEA was performed to more systematically correlate gene expression patterns with cellular functions and found that translation-related pathways were positively enriched in responders, while immune-related pathways were mainly suppressed in responders' tumor samples but activated in responders' immune samples, especially B cell-mediated immunity, immunoglobulin-mediated immunity, and immune cell receptor signaling pathways. In addition, not only T cells but also B cells and macrophages were activated, suggesting that other types of immune cells are involved in the response to ICI treatment, not just T cells. In particular, although it was not possible to determine from the pathway analysis what type of protein translation was upregulated, it is possible that immunoglobulin synthesis was upregulated in ICI responders,

given that B cell and immunoglobulin-mediated immunity pathways were activated in ICI responders. These results are consistent with previous studies that have identified B cell-mediated immunity as involved in the response to ICI treatment in other cancers[12,13], and that plasma cells are also involved in the response to ICI treatment in NSCLC[14].

To determine whether the functional changes in response to ICI treatment in each histologic subtype of NSCLC were similar to those in the entire NSCLC, we performed GSEA on each of the LUSC and LUAD samples. Even though the results were not identical to the GSEA results in the entire NSCLC sample, similar patterns were identified for immune-related pathways, confirming that these changes in immune function are not specific to any one histologic type.

In addition, we used NMF and WGCNA methods to identify gene modules associated with ICI treatment response and confirmed that these gene modules were associated with immune cells such as B cells, neutrophils, monocytes, and macrophages, as well as T cells, suggesting that immune cells other than T cells are also involved in ICI treatment response in the module analysis as well as in the functional analysis through GSEA.

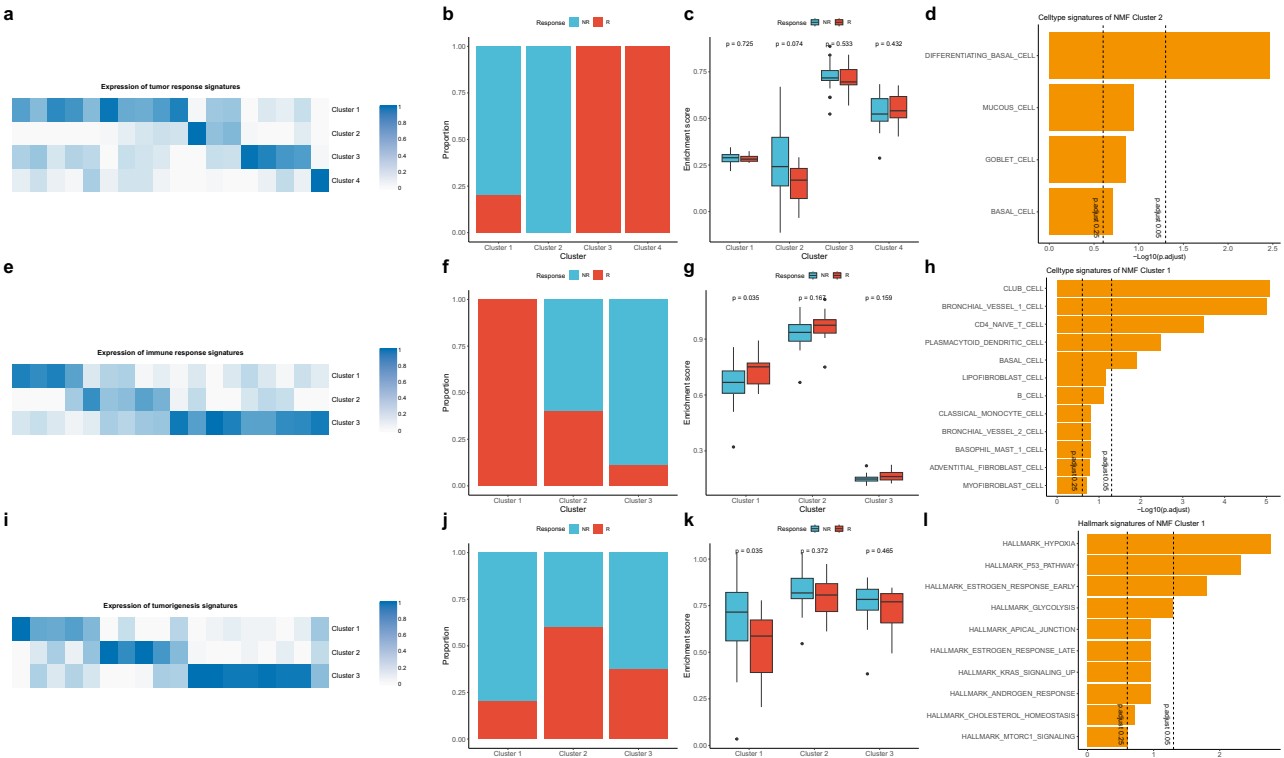

**Fig. 4 | Derivation of clusters by NMF and characteristics of each cluster. a** The heatmap of NSCLC tumor H-matrix and NMF-derived signature expression of tumor DEGs (tumor response signature). **b** Proportion of tumor responder and non-responder in each tumor response cluster. **c** Barplot showing ssGSEA enrichment score of tumor response signatures in the public dataset (GSE126044, GSE135222). The *p*-value between responder and non-responder in each cluster was measured using Kruskal–Wallis test. **d** Barplot showing ORA results using MSigDB cell type signature gene sets in tumor response cluster 2. The dashed line indicates an adjusted *P*-value of 0.25 and 0.05. **e** The heatmap of NSCLC immune H-matrix and NMF-derived signature expression of immune DEGs (Immune response signature). **f** Proportion of immune responder and non-responder in each immune response cluster. **g** Barplot showing ssGSEA enrichment score of immune response signatures in the public dataset (GSE126044, GSE135222). **h** Barplot showing ORA results using MSigDB celltype signature gene sets in immune response cluster 1. **i** The heatmap of NSCLC tumor H-matrix and NMF-derived signature expression of DEGs between tumor vs. immune samples (tumorigenesis signature). **j** Proportion of tumor responder and non-responder in each tumorigenesis clusters. **k** Barplot showing ssGSEA enrichment score of tumorigenesis signatures in the public dataset (GSE126044, GSE135222). **l** Barplot showing ORA results using MSigDB hallmark gene sets in tumorigenesis cluster 1. NR non-responder, R responder, NMF non-negative matrix factorization, ssGSEA single sample gene set enrichment analysis. Box plot components: The center line of the box represents the median, and the upper and lower limits of the box represent the upper and lower quantiles, respectively. Whiskers indicate a value of 1.5 times the interquartile range.

Recently, there have been several studies on the role of B cells in antitumor immune responses[15–17], and in this study, we focused on B cells, as B cell-mediated immune responses have been consistently found to be activated in responders to ICI treatment. We performed functional analysis using B cell subtype genes obtained from the literature review and validated with other datasets. In the Severance cohort, we found that all B cell subtypes were activated in responders to ICI treatment, and the changes were particularly pronounced in immune samples. In the validation dataset, we found a trend toward activation of all B cell subtypes in responders to immune checkpoint inhibitor treatment, although this was not significant across all B cell subtypes. We also found that higher expression of B cell immunity-related genes was associated with prolonged median survival in the validation dataset as well as in the Severance cohort.

Our study had several limitations. One limitation of this study is its relatively small sample size, which may limit the generalizability of the results. A larger patient cohort would provide more robust results and increase the statistical power of the analysis. In addition, our study only confirmed that B cells are involved in the ICI treatment response but did not identify the mechanisms by which B cells influence the ICI treatment response. Furthermore, this study may not have captured the full spectrum of responses to other immunotherapeutic agents or combined therapies.

In conclusion, this study provides comprehensive insights into the gene expression characteristics in response to immune checkpoint inhibitor

treatment in NSCLC. The identified genes and markers associated with B cell immune responses can be used as biomarkers to predict ICI treatment responses and select appropriate treatment candidates. Further studies are needed to investigate the mechanisms by which B cells influence the ICI treatment response in NSCLC, which may provide a new perspective on ICI therapy.

## Materials and methods
### Sample collection
Patients diagnosed with stage IV NSCLC and treated with PD-1/PD-L1 blockade (Nivolumab or Pembrolizumab) between March 2017 and December 2020 at Severance Hospital were included in this study. All patients agreed to participate in the lung cancer registry cohort, allowing the use of human-derived materials acquired during diagnostic research procedures. All patients provided written informed consent for inclusion in this cohort. This study adhered to the recommendations of the World Medical Association Declaration of Helsinki and was approved by the Institutional Review Board of Severance Hospital (IRB #4-2021-1747).

### GeoMx digital spatial transcriptomics
To compare differential gene expression in tumor and immune cells, analysis based on the NanoString GeoMx™ Digital Spatial Profiling (DSP) technology was performed (Fig. 1A). A tissue microarray with a 2 mm core

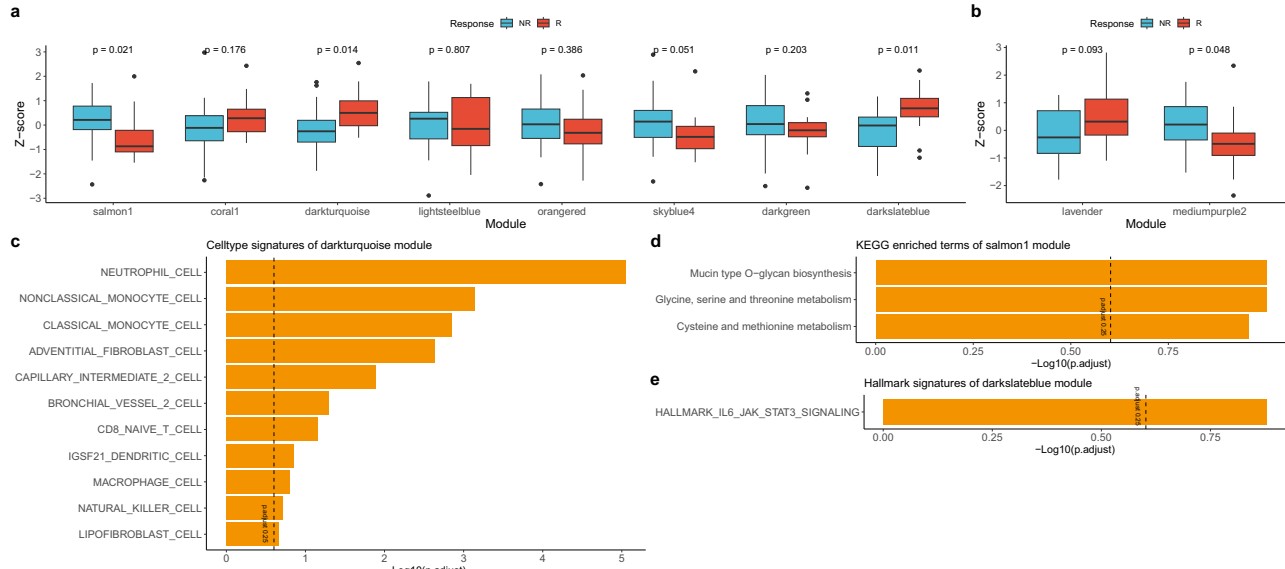

**Fig. 5 | Expression profiling and molecular characteristics of WGCNA modules in public datasets (GSE126044, GSE135222). a, b** Expression barplot of **a** tumor and **b** immune WGCNA modular genes by response in public datasets (*n* = 43). The *P*-value between the responder and non-responder in each module was measured using the Kruskal–Wallis test. **c** Barplot showing ORA results using MSigDB cell type signature gene sets in the dark turquoise module. **d** Dotplot showing ORA results using MSigDB KEGG gene sets in the salmon1 module. **e** Dotplot showing

ORA results using MSigDB hallmark gene sets in the dark slate blue module. The dashed line indicates an adjusted *P*-value of 0.25. NR non-responder, R responder. Box plot components: The center line of the box represents the median, and the upper and lower limits of the box represent the upper and lower quantiles, respectively. Whiskers indicate a value of 1.5 times the interquartile range.

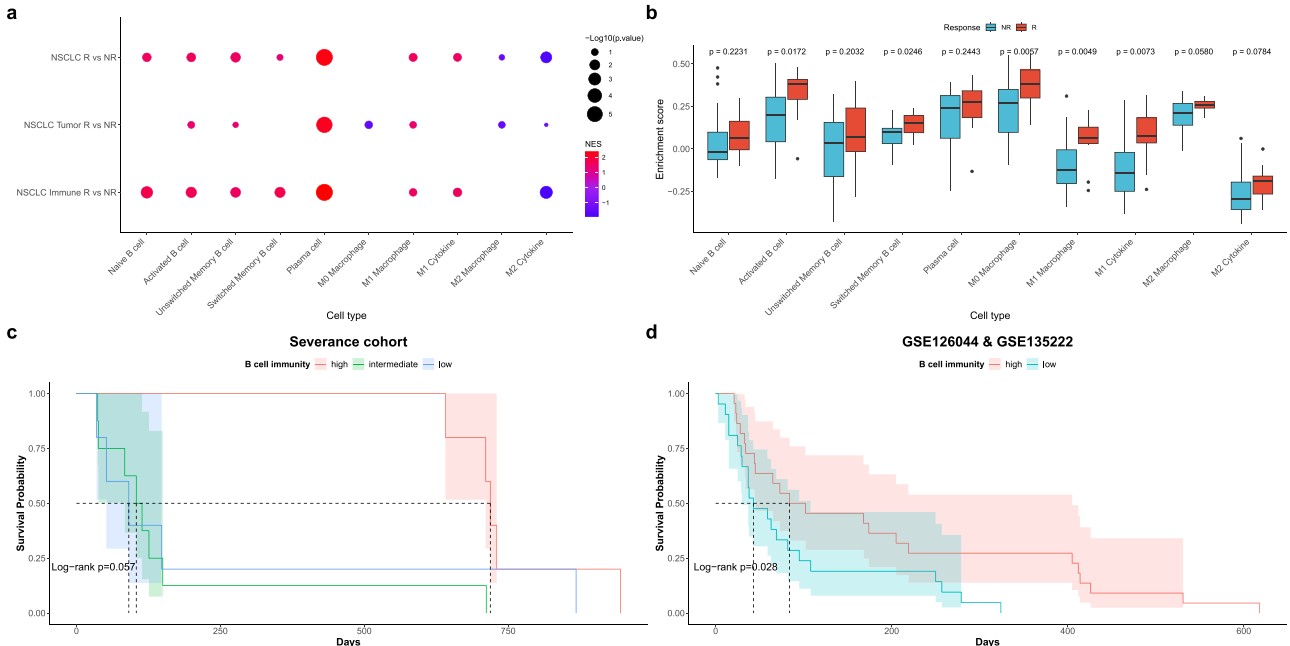

**Fig. 6 | Expression profiling and survival analysis of immune cell markers in severance NSCLC cohort and public datasets (GSE126044, GSE135222). a** Dotplot is showing GSEA results using B cell and macrophage subtype markers NSCLC responder and non-responder samples (*n* = 18). **b** Barplot shows ssGSEA enrichment score of B cell and macrophage markers in the public dataset (*n* = 43). The *p*-value between responder and non-responder in each module was measured

using Kruskal–Wallis test. **c, d** Kaplan–Meier plot of survival probability by B cell immunity in **c** severance NSCLC cohort (*n* = 18) and **d** public datasets (*n* = 43). NR non-responder, R responder. Box plot components: The center line of the box represents the median, and the upper and lower limits of the box represent the upper and lower quantiles, respectively. Whiskers indicate a value of 1.5 times the interquartile range.

diameter was constructed from formalin-fixed paraffin-embedded tumor tissues from eligible patients. Fixed formalin-fixed paraffin-embedded tissue sections of 5 μm thickness were mounted on charged slides. A representative region per sample was chosen as the region of interest (ROI) by a pulmonary pathologist (Fig. S1A). To segment areas of tumor and immune cells within

ROIs, multiplexed immunofluorescence staining was done by Cell DIVE™ technology (Leica Microsystems, Issaquah, USA) with morphologic markers as follows; anti-pan-cytokeratin for tumor cells (AE1/AE3, Novus, USA) and anti-CD45 for immune cells (D9M8I, Cell Signaling Technology, USA). Consequently, two areas of interest were selected per ROI. For each

area of interest, the GeoMx whole-transcriptome atlas (human RNA) profile was measured. From slide preparation to transcriptomic data acquisition, detailed methods were followed as described in a previous protocol by Merritt et al.[18]. After data collection, the reporter count conversion files were loaded into the GeoMx Digital Spatial Profiler analysis suite (V.2.4.2.2), where quality control and data scaling were performed. None of the ROIs met the quality control criteria. Subsequently, the data were scaled to the geometric mean of the number of nuclei and exported to R for further analysis.

## Transcriptome analysis

Downstream analysis of the transcriptome expression profiling obtained from each ROI was performed using R. Normalization of raw transcript counts, and differential gene expression (DEG) analysis was conducted using the limma-voom pipeline of the limma R package[19,20]. Genes with an adjusted $P$ value ($P_{adj}$) or nominal $P$ value ($P_{nom}$) $\leq 0.05$ were defined as DEGs. Data visualization was performed using the R packages Complexheatmap, ggplot2, and ggpubr.

## Functional enrichment analysis

For the functional analysis of transcriptome expression profiling, gene set enrichment analysis (GSEA) was performed using the clusterProfiler R package[21,22]. The gene sets used in GSEA were the hallmark gene set and cell type signature gene set from the MSigDB (Molecular Signatures Database) collection on the GSEA website (https://www.gsea-msigdb.org/gsea/index.jsp), and the GO BP (Gene Ontology Biologic Process) gene set and KEGG (Kyoto Encyclopedia of Genes and Genomes) gene set. To analyze immune cell subtypes, we conducted a literature review and identified B cell subtypes called naïve, activated, unswitched and switched memory B cells and plasma cells; macrophage subtypes called M1 macrophage and M2 macrophage and markers for cytokines corresponding to each macrophage subtype. These markers were used to perform GSEA and ssGSEA (single sample GSEA) for immune cell subtypes. GSEA and ssGSEA for immune cell subtypes were performed using these markers[23–26]. Significantly enriched pathways were defined as pathways with $P_{nom} \leq 0.05$ in the analysis of immune cell subtypes and $P_{adj} \leq 0.05$ for other analyses.

## Non-negative matrix factorization (NMF)

In addition to WGCNA, NMF was performed as an alternative method to identify gene modules associated with immune checkpoint blockade responses. The DEGs ($P_{adj} \leq 0.05$) between tumor and immune samples were defined as tumorigenesis signature if log2 fold change (LFC) $\geq 1$ and immune cell signature if LFC $\leq -1$. We also defined the DEGs ($P_{nom} \leq 0.05$, LFC $\geq 0.5$, or LFC $\leq -0.5$) between responder and non-responder for immune checkpoint blockade in tumor samples as tumor response signatures and the DEGs in immune samples as immune response signatures. NMF was performed on each signature, and the number of clusters was determined by the value that maximized the cophenetic correlation coefficient and dispersion coefficient. For each cluster, ssGSEA and ORA (over-representation analysis) was performed to identify the clusters associated with responders and non-responders and their functions.

## Weighted gene co-expression network analysis (WGCNA)

WGCNA was performed to identify gene modules associated with immune checkpoint blockade responses. First, the raw transcript counts of tumor and immune samples were transformed to normalized log2 CPM (Count per million). Genes with log2 CPM $\geq 0.1$ were used for downstream analysis. To achieve a scale-free topology fit index of 0.9 or higher for the entire sample and for subgroup analysis, we set the soft thresholding power ($\beta$ value) as 12 for entire NSCLC samples and NSCLC tumor samples and 10 for NSCLC immune samples. Module detection was conducted with a minModuleSize of 30 and merge-CutHeight of 0.25[27]. Only modules with a $P$-value below 0.05 were used for further analysis.

## Survival analysis

B cell signature gene expression for each tumor and immune tissue was obtained, and each tumor and immune tissue was divided into B cell signature high and low groups according to B cell signature gene expression. Then, to divide patients into groups according to B cell signature expression, patients were classified into the B cell signature high group if both their tumor and immune tissues were in the B cell signature high group, into the intermediate group if either was in the high group, and into the low group if both tissues were in the low group. Kaplan–Meier survival analysis was performed to compare progression-free survival in each group. In the validation dataset, patients were categorized into B cell signature high and low groups according to B cell signature gene expression, and Kaplan–Meier survival analysis was performed. The significance of the survival analysis was expressed as a log-rank $P$ value.

## Statistics and reproducibility

All statistical analyses were performed using R software (version 4.3.3). RNA seq data analysis was performed based on the limma-voom pipeline of the limma R package and defined as DEGs based on $P_{adj}$ or $P_{nom} \leq 0.05$. GSEA for transcriptome expression profiling was performed using the Clusterprofiler R package, and pathways with $P_{adj} \leq 0.05$ were defined as significantly enriched pathways. For the analysis of B cell and macrophage subtypes only, pathways with $P_{nom} \leq 0.05$ were defined as significantly enriched pathways. The WGCNA package was used to perform WGCNA analysis on genes with a normalized log2 CPM > 0.1, and modules with $P_{nom} \leq 0.05$ were defined as significant modules and used for downstream analysis. Kaplan–Meier survival analysis was performed on NSCLC patients (Severance cohort and validation dataset) divided into B cell immunity high, intermediate, and low groups according to B cell signature gene expression, and the significance of survival analysis was expressed as a log-rank $P$ value. The sample size of each group used in the analysis is indicated in the figure legend.

## Reporting summary

Further information on research design is available in the Nature Portfolio Reporting Summary linked to this article.

## Data availability

The anonymized datasets and additional documents used in this study can be made available upon reasonable request from the corresponding author with institutional approval. All human-derived materials from the Yonsei University College of Medicine, Severance Hospital, were used for research purposes for the current study and are not publicly available. The spatial transcriptomics data analyzed in this article are publicly available in the K-BDS database (BioProject ID: temp-grp-2-1720057798276).

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

## Acknowledgements
This research was supported by the National Research Foundation of Korea (NRF) grant funded by the Korea government (MSIT) (No. 2022R1F1A1068789), the Basic Science Research Program of the National Research Foundation of Korea, funded by the Ministry of Education (No. 2020R1C1C1004999), GIST Research Institute IIBR grants funded by the GIST in 2023, and the KHIDI-AZ Diabetes Research Program. This research was also supported by a grant from the MD-PhD/Medical Scientist Training Program of the Korea Health Industry Development Institute (KHIDI), funded by the Ministry of Health and Welfare, Republic of Korea.

## Author contributions
S.K., S.H.L., and C.M.O. conceived and designed this study. S.H.Y, H.H.K., and S.K. collected. S.H.Y. collected and analyzed clinical data. J.K., G.J., Y.K., R.P., and C.M.O. performed spatiotemporal transcriptomic analysis and other gene tests. J.K. and S.H.Y wrote and revised the manuscript. S.H.Y, J.K., S.K., C.M.O, and S.H.L. had access to all the data. All the authors contributed to and approved the final manuscript.

## Competing interests
The authors declare no competing interests.
