## [Transparent Peer Review file · Communications Biology]

Spatial profiling of non-small cell lung cancer provides insights into tumorigenesis and immunotherapy response

Corresponding Author: Professor Chang-Myung Oh

Version 0:

Reviewer comments:

Reviewer #1

(Remarks to the Author)

Kim, et al., described the spatial profiling of non-small cell lung cancer (NSCLC) and discovered sets of genes related to tumorigenesis and immunotherapy response. These gene signatures may be useful in the further study of NSCLC. Overall, this paper is a standard spatial transcriptomics study. Thus, the biological findings are important. A few suggestions are listed below:

1. Are these genes found for the first time or have been described before by using other methods?
2. Is there any way to validate the findings? Maybe a bigger sample size for the genes found?
3. Is there sampling bias when using DSP? How does it compare to other spatial transcriptomic methods?
4. There are very detailed descriptions of the results, but the conclusions are vague. For example, for the genes that are related to immunotherapy response, can they be used as biomarkers? How to validate this kind of findings?

Reviewer #2

(Remarks to the Author)

In the manuscript entitled "Spatial profiling of non-small cell lung cancer: Unveiling tumorigenesis and immunotherapy response", Kim et. al, aimed to identify the molecular features in NSCLC patients after immunotherapy with different response based on spatial transcriptome (ST) data. It will be better to address some issues as follows.

Major comments

1. Understanding the unique molecular features or the mechanism of non-response or response to immunotherapy is an essential topic. However, it is hard to learn the answer from current version of the manuscript. Besides, compared to other data, ST may provide new insights into cell communications and niches, in the current version, the author's analysis should highlight the new knowledge found comparing previous bulk rna-seq data and single-cell data.
2. The authors have done some DEGs between response and non-response groups, and a few genes were observed. Which cell types play the key role, and how they regulate others needs to be further analyzed.

Minor comments:

1. The conclusion of "ApoE, IGKC, LYZ were found to be commonly upregulated in both NSCLC responders and non-responders compared with normal tissues" seems to contradict Figure 1C.
2. Please provide more evidence when saying "DEG analysis identified genes specific to LUSC and LUAD and functional enrichment analyses suggested their involvement in cell adhesion, migration, and immune response-related processes.", because in the article only the pathway analysis of LUSC is related to migration.
3. According to tumor/normal, LUSC /LUAD, responder/non-responder and so on, multiple data groups were divided before further analysis and found the DEGs. Then, combining the DEGs functional enrichment analysis was done from multiple perspectives. Although differences in pathways have been found, it is still not very clear what factors lead to differences in patients' responses to immune checkpoint therapy, please comment.
4. In the analysis of Figure 5, multiple genes associated with a favorable response to Some abbreviations did not follow with full form phrases when they first appeared like "RCC", "HPA" and "MS". Besides, the inconsistent abbreviations like "ICI /IC" need to be revised.
5. It is misleading to place a t-SNE plot in the data analysis section of Figure 1A because there is no plot of cell clustering in

the study.

6. The figure legend of Figure 1 is confusing, e.g. "tumor vs. immune cells".

7. Grouping information should be given on figures like Figure 5A, Figure S2. The Grouping information of "Response tumor vs nonresponse tumor" for example in Figure 2B and Figure S3C should be unified.

8. Please give a detailed description about the methods of performing QC and data scaling

9. How did you process the data when doing the analysis related to the group of responder vs. non-responder (tumor/immune fold change).

10. In Figure 5C, what was the standard for judging the significance of module of MEdarkorange ($R^2=0.97$, $P=0.001$) when saying "In LUAD, no module showed significant MS (Figure 5C)".

Author Rebuttal letter:

=====
Responses to Reviewer #1's Comments
=====

Kim, et al., described the spatial profiling of non-small cell lung cancer (NSCLC) and discovered sets of genes related to tumorigenesis and immunotherapy response. These gene signatures may be useful in the further study of NSCLC. Overall, this paper is a standard spatial transcriptomics study. Thus, the biological findings are important. A few suggestions are list below:

[Comment #1]

1. Are these genes found for the first time or have been described before by using other methods?

[Response #1]

We are grateful for Reviewer #1's comment. There have been recent studies showing a role for B cell immunity in immune checkpoint response and T cell activation in other cancers (<https://doi.org/10.1016/j.cell.2019.10.028>). In addition, studies have shown that plasma cells play an important role in the immune checkpoint response in NSCLC (<https://doi.org/10.1016/j.ccell.2022.02.002>). Therefore, we wanted to investigate the role of B cell immunity on the immune checkpoint response in NSCLC, and identified B cell immunity-related genes identified in the literature review that play an important role in the immune checkpoint response.

[Comment #2]

2. Is there any way to validate the findings? Maybe a bigger sample size for the genes found?

[Response #2]

Due to the small sample size of this study, we decided that validation was necessary, so we performed validation using public data (GSE126044, GSE135222). The validation performed on public data confirmed that B cell immunity is associated with immune checkpoint response. We added this validation result in figure 5 as follows:

[Comment #3]

3. Is there sampling bias when using DSP? How does it compare to other spatial transcriptomic methods?

[Response #3]

Thank you for your nice comment. Nanostring GeoMx DSP is a technology that allows us to select a region of interest (ROI) in a sample and view transcriptome expression per ROI. To minimize sampling bias, after staining tumor and immune cell markers using IHC, our pathologists distinguish tumor immune tissue based on this. In addition, ESTIMATE analysis was performed on the transcriptome data, and it was possible to re-confirm that the tumor and immune samples were well divided.

[Comment #4]

There are very detailed descriptions of the results, but the conclusions are vague. For example, for the genes that are related to immunotherapy response, can they be used as biomarkers? How to validate this kind of findings?

[Response #4]

Thank you for your nice comment. As you commented, the conclusions are vague. In the previous analysis, we performed transcriptome analysis using the DESeq2 R package. During the revision process, we realized that

DESeq2 does not allow for experimental designs that take into account matched samples. Therefore, to improve our analysis, we re-analyzed a significant part of the previous analysis using the limma-voom R package. We then identified several markers of B cell immunity through a literature review and validated that they were associated with prognosis (Figure 6 and supplementary Table 2). We added some comments about this package in the method part as follows:

Downstream analysis of the transcriptome expression profiling obtained from each ROI was performed using R language. Normalization of raw transcript counts and differential gene expression (DEG) analysis was conducted using the limma-voom pipeline of the limma R package. Genes with an adjusted P value (Padj) or nominal P value (Pnom) ≤ 0.05 were defined as DEGs. Data visualization was performed using the R packages Complexheatmap, ggplot2, and ggpubr.

And we added new result in Figure 6 as follows:

=====
Responses to Reviewer #2's Comments
=====

In the manuscript entitled "Spatial profiling of non-small cell lung cancer: Unveiling tumorigenesis and immunotherapy response", Kim et. al, aimed to identify the molecular features in NSCLC patients after immunotherapy with different response based on spatial transcriptome (ST) data. It will be better to address some issues as follows.

[Comment #1]

Understanding the unique molecular features or the mechanism of non-response or response to immunotherapy is an essential topic. However, it is hard to learn the answer from current version of the manuscript. Besides, compared to other data, ST may provide new insights into cell communications and niches, in the current version, the author's analysis should highlight the new knowledge found comparing previous bulk RNA-seq data and single-cell data.

[Response #1]

Based on this point, we reperform most of our analysis and made new figures based on this analysis. There have been recent studies implicating B cell immunity in T cell activation in other type of cancer. Furthermore, in NSCLC, plasma cells have been shown to play an important role in immune checkpoint response. In this study, we found a lot of B cell infiltration & activation in immune cell rich areas, but also some B cell infiltration & activation in tumor tissue. New Figure 6 shows the significant markers of immune cell response and validation result of this new analysis.

And we added new comments this new knowledge in the discussion part as follows:

We identified gene expression patterns in response to treatment in NSCLC samples and found that translation-related genes and some immune-related genes were, while some other immune-related genes were downregulated upregulated in ICI responders. GSEA was performed to more systematically correlate gene expression patterns with cellular functions and found that translation-related pathways were positively enriched in responders, while immune-related pathways were mainly suppressed in responders' tumor samples but activated in responders' immune samples, especially B cell-mediated immunity, immunoglobulin-mediated immunity, and immune cell receptor signaling pathways. In addition, not only T cells, but also B cells and macrophages were activated, suggesting that other types of immune cells are involved in the response to ICI treatment, not just T cells. Although it was not possible to determine from the pathway analysis what type of protein translation was upregulated, it is possible that immunoglobulin synthesis was upregulated in ICI responders, given that B cell and immunoglobulin-mediated immunity pathways were activated ICI responders. These results are consistent with previous studies that have identified B cell-mediated immunity are involved in the response to ICI treatment in other cancers, and that plasma cells are also involved in the response to ICI treatment in NSCLC.

[Comment #2]

2. The authors have done some DEGs between response and non-response groups, and a few genes were observed. Which cell types play the key role, and how they regulate others needs to be further analyzed.

[Response #2]

During the new analysis, we identified several genes related to B cell immunity through a literature review and confirmed that these genes can be used to predict immune checkpoint response.

We added some comments about this in the result part and discussion part as follows:

Result part:

In the above analysis, we found that B cell and M1 macrophage function varied in response to immune checkpoint inhibitor treatment, and to determine if this actually affected patient survival, we performed Kaplan-Meier survival analysis and found that median survival of high B cell immunity group (749 days) was higher than intermediate or low B cell immunity group (171 and 239 days) in the Severance cohort, although it didn't reach statistical significance ($P = 0.057$), and in the validation dataset, median survival was significantly increased in the high B cell immunity group (189 days) than low B cell immunity group (89 days).

Discussion part:

The identified genes and markers associated with B cell immune response can be used as biomarkers to predict ICI treatment response and select appropriated treatment candidates. Further studies are needed to investigate the mechanisms by which B cells influence ICI treatment response in NSCLC, which may provide a new perspective on ICI therapy.

Regarding the regulation, we didn't perform functional analysis to elucidate the exact mechanism, so we added some comments about this limitation as follows:

In addition, our study only confirmed that B cells are involved in ICI treatment response but did not identify the mechanisms by which B cells influence ICI treatment response. Furthermore, this study may not have captured the full spectrum of responses to other immunotherapeutic agents or combined therapies.

[Minor Comments]

1. The conclusion of ApoE, IGKC, LYZ were found to be commonly upregulated in both NSCLC responders and non-responders compared with normal tissues seems to contradict Figure 1C.

[Response #1]

Thank you for your comment. We have reanalyzed our data and as a result, those data have been removed from the figures. Supplementary Table 1 show the DEGs between groups. In addition, we made new figures as follows:

Figure 1

Figure 1. Expression profiling and molecular characteristics of NSCLC transcriptome by response. (B - D) Volcano plot of DEGs between responder and non-responder of (B) all NSCLC ($n = 18$), (C) tumor and (D) immune samples. Only genes with nominal p-value ≤ 0.05 were considered DEGs.

2. Please provide more evidence when saying "DEG analysis identified genes specific to LUSC and LUAD and functional enrichment analyses suggested their involvement in cell adhesion, migration, and immune response-related processes," because in the article only the pathway analysis of LUSC is related to migration.

[Response #2]

Thank you for your comment. We have reanalyzed our data and as a result, those data have been removed. We performed GSEA and added new results in the results part as follows:

GSEA was performed to identify functional differences between immune checkpoint inhibitor responders and non-responders in LUSC. GSEA using MSigDB Hallmark and GO BP terms revealed positive enrichment of pathways related to glucose metabolism and proliferation, ribosome, antigen processing and presentation in the responders, as well as in the whole NSCLC samples (Figure 2D - 2E). GSEA using KEGG terms also confirmed the positive enrichment of anabolism-related pathways such as translation, ribosome biogenesis, peptide, amide biosynthesis, and other immune-related pathways, and the positive enrichment of antigen processing and presentation-related pathways (Figure 2F).

In LUAD, as in entire NSCLC and LUSC samples, we performed functional analysis to identify differences between ICI responders and non-responders (Figure 3D - figure 3G). GSEA using MSigDB Hallmark terms confirmed that LUAD ICI responders were positively enriched for interferon gamma response compared to non-responders. In the metabolism-related pathways, only fatty acid metabolism was positively enriched, while proliferation pathways were mostly negatively enriched. GSEA using GO BP terms revealed positive enrichment of the translation-related ribosome pathway, similar to the whole NSCLC and LUSC samples. Glycolysis/gluconeogenesis pathway was found to be positively enriched only in responders of LUAD tumor

samples. Analysis using KEGG terms showed positive enrichment of translation and amide biosynthesis pathway in entire LUAD samples and LUAD immune samples, as well as B cell mediated immunity-related pathway and phagocytosis-related pathway in entire NSCLC samples and LUSC samples. GSEA using the cell type signatures gene set confirmed the positive enrichment of macrophages.

We also add new figures as follows:

Figure 2

Figure 3

3. According to tumor/normal, LUSC/LUAD, responder/non-responder and so on, multiple data groups were divided before further analysis and found the DEGs. Then, combining the DEGs functional enrichment analysis was done from multiple perspectives. Although differences in pathways have been found, it is still not very clear what factors lead to differences in patients' responses to immune checkpoint therapy, please comment.

[Response #3]

Thank you for your comment. To improve our data clearly, we re-analyzed our data. The new analysis confirmed that T cell and B cell immunity is activated in the responder in NSCLC patients treated with immune checkpoint inhibitors. We made new figure and added comments about this in the discussion part as follows:

Figure 6

In addition, we used NMF and WGCNA methods to identify gene modules associated with ICI treatment response and confirmed that these gene modules were associated with immune cells such as B cells, neutrophils, monocytes, and macrophages, as well as T cells, suggesting that immune cells other than T cells are also involved in ICI treatment response in the module analysis as well as in the functional analysis through GSEA.

4. In the analysis of Figure 5, multiple genes associated with a favorable response to Some abbreviations did not follow with full form phrases when they first appeared like "RCC", "HPA" and "MS". Besides, the inconsistent abbreviations like "ICI /IC" need to be revised.

[Response #4]

Thank you for your comment. We fixed a potentially confusing section in the new version of the manuscript. 5. It is misleading to place a t-SNE plot in the data analysis section of Figure 1A because there is no plot of cell clustering in the study.

[Response #5]

Thank you for your comment. We fixed a potentially confusing section in the new version of the manuscript.

6. The figure legend of Figure 1 is confusing, e.g. tumor vs. immune cells.

[Response #6]

Thank you for your comment. We changed tumor/immune cell to tumor/immune sample, where each refers to a tumor cell-rich region and an immune cell-rich region in each patient's lung cancer tissue.

7. Grouping information should be given on figures like Figure 5A, Figure S2. The Grouping information of Response tumor vs nonresponse tumor for example in Figure 2B and Figure S3C should be unified.

[Response #7]

Thank you for your comment. We fixed a potentially confusing section in the new version of the manuscript.

8. Please give a detailed description about the methods of performing QC and data scaling

[Response #8]

QC and data scaling methods are further described in Method part as follows:

After data collection, the RCC files were loaded into the GeoMx DSP analysis suite (V.2.4.2.2), where quality control (QC) and data scaling were performed. None of the ROIs met the QC criteria. Subsequently, data were scaled to the geometric mean of the number of nuclei and exported to R for further analysis.

9. How did you process the data when doing the analysis related to the group of responder vs. non-responder (tumor/immune fold change).

[Response #9]

Thank you for your comments. Following the reviewer's comments, we performed a re-analysis to compare gene expression of tumor samples with gene expression of normal tissues using the limma-voom package. The original purpose of this analysis was to compare transcript expression between responder and non-responder tumor samples. Therefore, we excluded this responder vs. non-responder (tumor/immune fold change) comparison data to minimize confounding results.

10. In Figure 5C, what was the standard for judging the significance of module of MEdarkorange

($R^2=0.97$, $P=0.001$) when saying ln LUAD, no module showed significant MS (Figure 5C).

[Response #10]

Thank you for your comments. The new analysis changed many parts of the figure, so the new figure does not include the old figure 5C. We comment about this in the result part and added new Figure 5 as follows:

To determine whether these modules were enriched differently between responders and non-responders, we performed ssGSEA and found that among the modules identified in NSCLC tumor samples, salmon1 ($P = 0.021$), darkturquoise ($P = 0.014$), and darkslateblue module ($P = 0.011$) were found to have significantly different enrichment scores in responders and non-responders, with the salmon1 module having a higher enrichment score in non-responders, and the darkturquoise and darkslateblue modules having a higher enrichment score in responders (Figure 5A - 5B). To determine the function of each of these three modules, we performed ORA on each of the modular genes and found that neutrophil, monocyte, macrophage and CD8 T cell in the darkturquoise module, O-glycan and several amino acid metabolism-related pathways in the salmon1 module, and IL6/JAK/STAT3 pathway in the darkslateblue module were enriched (Figure 5C - 5E).

Figure 5. Expression profiling and molecular characteristics of WGCNA modules in public datasets (GSE126044, GSE135222). (A)

- B) Expression barplot of (A) tumor and (B) immune WGCNA modular genes by response in public datasets ($n = 43$). The p-value between responder and non-responder in each module was measured using Kruskal-Wallis test. (C) Barplot showing ORA results using MSigDB celltype signature gene sets in darkturquoise module. (D) Dotplot showing ORA results using MSigDB KEGG gene sets in salmon1 module. (E) Dotplot showing ORA results using MSigDB hallmark gene sets in darkslateblue module. The dashed line indicates an adjusted p-value 0.25. NR non-responder, R responder.

Version 1:

Reviewer comments:

Reviewer #1

(Remarks to the Author)

The authors have addressed my concerns fairly well. I don't have further comments.

Reviewer #2

(Remarks to the Author)

In the revised version, the authors have added some reanalysis results between non-responders and responders. However, the results are descriptive and it is hard to know what is the new finding in this study. Besides, the quality of the figures is weak.
